# Perspectives on linkage to care for patients diagnosed with HIV: A qualitative study at a rural health center in South Western Uganda

**Mark Opio[1], Florence Akello[1], Doreen Kagina Twongyeirwe[1], David Opio[1], Juliet Aceng[2], Jane Kasozi Namagga[1], Jerome Kahuma Kabakyenga[3]\***

1 Department of Nursing, Mbarara University of Science and Technology, Mbarara, Uganda, 2 Department of Medical Laboratory Science, Mbarara University of Science and Technology, Mbarara, Uganda, 3 Maternal Newborn and Child Health Institute, Mbarara University of Science and Technology, Mbarara, Uganda

\* jkabakyenga@must.ac.ug

**Data Availability Statement:** All relevant data are within the paper and its Supporting Information files.

## Abstract

Linkage to care for newly diagnosed human immunodeficiency virus (HIV) patients is important to ensure that patients have good access to care. However, there is little information about factors influencing linkage to care for HIV patients. We aimed to identify existing measures in place that promote linkage to care and to explore facilitators and barriers to linkage to care for clients diagnosed with HIV/acquired immune deficiency syndrome at a rural health center in Uganda. This descriptive qualitative study enrolled 33 purposively selected participants who included expert clients, linkage facilitators, heads of families with people living with HIV, and health workers. Data were collected using in-depth interviews that were audio-recorded, transcribed, and translated. The data were manually analyzed to generate themes. The following four themes were generated: 1) availability of services that include counseling, testing, treatment, follow-up, referral, outreach activities, and support systems. 2) Barriers to linkage to care were at the individual, health facility, and community levels. Individual-level barriers were socioeconomic status, high transport costs, fear of adverse drug effects, fear of broken relationships, and denial of positive results or treatment, while health facility barriers were reported to be long waiting time, negative staff attitude, and drug stock outs. Community barriers were mostly due to stigma experienced by HIV clients, resulting in discrimination by community members. 3) Facilitators to linkage to care were positive staff attitudes, access to information, fear of death, and support from others. 4) Suggestions for improving service delivery were shortening waiting time, integrating HIV services, increasing staff numbers, and intensifying outreaches. Our findings highlight the importance of stakeholder involvement in linkage to care. Access and linkage to care are positively and negatively influenced at the individual, community, and health facility levels. However, integration of HIV services and intensifying outreaches are key to improving linkage to care.

**Funding:** MO, DKT, JA, DO, FA. Research reported in this publication was supported by the Fogarty International Center (U.S. Department of State's Office of the U.S. Global AIDS Coordinator and Health Diplomacy [S/GAC] and the President's Emergency Plan for AIDS Relief [PEPFAR]) of the National Institutes of Health under Award Number R25TW011210. The content is solely the responsibility of the authors and does not necessarily represent the official views of the National Institutes of Health".

**Competing interests:** The authors have declared that no competing interests exist.

**Abbreviations:** ART, Anti-retroviral therapy; ARVs, Anti-retroviral drugs; COVID-19, Coronavirus disease; SOPs, Standard operating procedures; HIV/AIDS, Human immunodeficiency virus/acquired immune deficiency syndrome; HTS, HIV-testing services; MUST, Mbarara University of Science and Technology; OPD, Outpatient department; PLHIV, People living with HIV; MUREC, Mbarara University of Science and Technology Research Ethics Committee; UNAIDS, Joint United Nations Programme on HIV/AIDS; UNCST, Uganda National Council for Science and Technology.

## Introduction

According to the Joint United Nations Programme on human immunodeficiency virus–acquired immune deficiency syndrome (UNAIDS), globally in 2020, there were 37.7 million people living with HIV (PLHIV), 1.5 million people got infected with HIV, and there were 680,000 deaths due to AIDS-related illness. Although anti-retroviral drugs are generally available, only 73% of PLHIV were on treatment in 2020 [1]. Sub-Saharan Africa is the region that was worst affected in carrying the major disease burden and has 460,000 HIV/AIDS-related deaths in 2020, accounting for two-thirds of the global toll [1]. In 2020, Uganda, one of the countries badly affected by HIV/AIDS, had 1.3 million PLHIV and was projected to have had 22,000 deaths in PLHIV in the same year [2].

Linkage to care for HIV-positive patients is a process of enrolling and ensuring that all HIV-positive patients have access to antiretroviral therapy (ART) and care services [3]. Linkage to care starts with HIV-testing services and is followed by post-test counseling, clinical evaluation, ART initiation, and early support up to the first follow-up visit [4]. Linkage to care has also been described as a pathway from HIV diagnosis to initial engagement with HIV care and treatment [5]. When linkage to different care points becomes inadequate, it leads to poor adherence, high viral load, and frequent opportunistic infections. This will have an impact on the quality of life of individuals and the general well-being of the community. The Joint UNAIDS recommends that a person diagnosed with HIV/AIDS should have immediate treatment initiation [6]. However, the start of ART is not the definition of linkage to care; rather, it is a key to part of the UNAIDS 90-90-90 targets for the HIV care continuum worldwide [7, 8]. The UNAIDS 90-90-90 targets were set to achieve the following: 90% of PLHIV knew their status; 90% of PLHIV who knew their status started and sustained on ART, and 90% of those on ART achieved viral load suppression [5]. Despite the UNAIDS strategies being put in place that require 90% of HIV-positive clients to receive sustained ART and other services, this has not been sufficiently achieved as reported in literature [9].

According to UNAIDS, in 2020, 84% of PLHIV knew their status, 73% were receiving treatment, and 66% had viral suppression [1]. Delayed linkage to care services in HIV-positive patients is a major barrier to treatment to reduce HIV transmission, thereby leading to increased disease burden in the community [10]. In sub-Saharan Africa, HIV remains a major disease burden, with 460 000 deaths associated with HIV in 2020 [1]. A study conducted in Ghana showed that linkage to care for HIV-positive patients remained poor at only 54% in 2019, whereas in a study conducted in South Africa, linkage to care for HIV-positive patients remained a problem, with only 27% linked by 2015 [4, 11]. In East Africa, studies on linkage to care conducted in Ifakara and Mbeya in Tanzania showed that only approximately 23% and 78% of HIV patients had access to linkage to care services, respectively, indicating that linkage to care is still a challenge [12, 13]. In 2018, a study on linkage to care conducted in 20 facilities in the central region of Uganda showed that only 53% of the HIV-positive clients had access to linkage to care within one month after testing [10] and a study conducted in one of the health facilities in Kabale District in South Western Uganda reported that 67% of HIV-positive patients had access to linkage to HIV care services. Similarly, a country progress report (2016–2017) on access to HIV services in one of the health facilities in Kabale district in South Western Uganda reported that 67% patients had access to linkage to care [14]. Rates of linkage to HIV care from various facilities are indicative of a challenging situation encountered by implementation programs.

In the Kinoni Health center in South Western Uganda, the quarterly report from April to June 2019 indicated that 84% of HIV-positive clients had linkage to HIV care services (Kinoni Health Center Management Information Services records, 2018/2019), which was below the UNAIDS target of 90%. This could indicate that while improvements occur in HIV testing services (HTS),

linkage to care remains below the targeted levels. Therefore, the present study aimed to identify the perspectives of health care providers and caretakers of PLHIV on existing measures in place that promote linkage to care, explore facilitators and barriers to linkage to care for clients diagnosed with HIV at Kinoni Health Center, Rwampara District, South Western Uganda.

## Materials and methods

### Study design

This was a descriptive qualitative study involving in-depth interviews. Data were collected between July and August of 2020.

### Study setting

The study was conducted at Kinoni Health center located in Rwampara District, located 26 km from Mbarara City. The Kinoni Health center, a government-owned health care facility, is designated as level IV according to the national health system structure [15, 16]. The health center offers both outpatient and inpatient services and serves a catchment area comprising 3 sub-counties, with a 2020 projected population of 66,000 [17]. The health center has an active HIV clinic with slightly more than 1100 registered active clients as of August 2020.

### Study population

The population included expert clients, linkage facilitators, health workers, and heads of families caring for PLHIV. Expert clients are PLHIV who are stable on ART but have undertaken training and are willing to serve fellow PLHIV [18, 19]. Linkage facilitators are community health workers who are selected by health facilities or implementing partners specifically to link HIV clients to care and community-based HIV services [20].

### Sample size

The sample size was achieved after reaching saturation on interviewing 33 participants. Data saturation was reached when no new information was found, whereby the researcher was able to derive categories, sub-themes, and themes [21].

### Sampling procedure

Participants were purposively selected for the study. These people were selected based on their experience and knowledge of interacting and working with HIV/AIDS patients, and for expert clients, they served a dual role as patients and peer educators.

### Data collection procedure

Data were collected using in-depth interview guides. Interview guides were used to guide the in-depth interviews with expert clients, linkage facilitators, health workers, and heads of families with PLHIV. The development of the in-depth interview guides was informed by three research questions: 1) What are the measures in place that link HIV clients to care? 2) What are the barriers and facilitators of linkage to care? 3) What are the possible solutions to improve linkage to care for HIV clients? The interview guides captured sociodemographic data, including age, sex, religion, occupation, and education level. The main body of the interview guides had open-ended questions focusing on the three research questions (measures in place, barriers, and facilitators to linkage, suggestions for improvement of services for linkage to care) designed for the specific groups of participants (S1 File).

The interview guide for health workers and linkage facilitators was designed in English. The interview guides for expert clients and heads of families with PLHIV were initially prepared in English and then translated to Runyankole/Rukiga, a local language spoken by the majority of the population in South Western Uganda. They were back-translated to English by individuals with experience in the translation of health-related information and were recognized by the local research ethics committee of the Mbarara University of Science and Technology Research Ethics Committee (MUREC). The back translation was used to maintain flow of information so that the translated questions reflected the original meaning [22]. The interview guides were approved by the MUREC.

The interview guides were pretested in a health facility in a neighboring district of Isingiro, and adjustments were made accordingly. Eligible participants were enrolled after obtaining written informed consent. The majority of the interviews were conducted at the health center, and only five interviews with heads of families with PLHIV living in a radius of approximately 5 km were conducted at their homesteads. All interviews were audio-recorded and lasted for 30–70 min. The data collected in the local language were transcribed and the transcripts translated to English, while the data collected in English were transcribed directly.

## Data analysis

Data were analyzed using thematic analysis [23]. The analysis followed a six-step approach, including familiarization, coding, generating, reviewing, defining and naming, and writing up themes found within the data. We listened and re-listened to audiotapes, read and reread notes, recalled observations, and repeated until the data were fully understood. The transcribed and translated data were crosschecked by the research team, which allowed a condensed overview of the main points and common meanings that recurred throughout the data. After reading the text, we collated all the data into groups identified by codes and categorized them to allow themes to emerge. The generation of themes was guided by theoretical constructs derived from the Health Belief Model [24, 25] which included perceived susceptibility, severity, health motivation, perceived benefits, barriers, and cue to action, and the ecological model [26] that focuses on the structural levels of the individual, community, and health care system. When an individual is diagnosed with HIV, the decision to be linked to care is affected by the person's perceptions, community, and health care system structural factors.

## Ethical considerations

The study was approved by the Mbarara University of Science and Technology Research Ethics Committee (MUREC 27/01-20) and registered by the Uganda National Council for Science and Technology (RESCLER/01). Permission to conduct the study was obtained from local district leaders. Participants provided written informed consent, and interviews were conducted in convenient places at the health facility and community.

## Results

The study findings are presented in two sub-sections: sociodemographic characteristics of the participants and the themes generated from participants' perspectives on measures in place, barriers, and facilitators of linkage to care for patients diagnosed with HIV.

### Sociodemographic characteristics of the participants

Regarding sociodemographic characteristics of the participants, 35 participants were approached. Two refused to participate: one was a health worker who refused due to lack of

time while the other was a head of family with PLHIV who claimed that interviewers were in the money-making business. We conducted in-depth interviews with 33 participants, including expert clients (10), linkage facilitators (4), heads of families of PLHIV (12), and health workers (7). Their ages ranged from 22 to 81 years, with a mean age of 46.7 years. The majority of the participants were female (66.7%), Christian (84.9%), attained primary/secondary-level education (60.6%), and were subsistence farmers (39.4%) or self-employed 27.3% (Table 1).

## Participants' perspectives on linkage to care for patients diagnosed with HIV

We identified four themes that were categorized as availability of services, barriers to linkage of care, facilitators to linkage of care, and suggestions for improving service delivery, as described below.

**Theme 1: Availability of services.** This theme emerged out of five sub-themes: testing services, counseling services, treatment services, follow-up, and support. Most of the participants reported receiving good services at the health facility, as illustrated in the following excerpts:

> "At the facility we had no problem because the health workers acted fast. The health workers are welcoming, drugs are available, and there are counselors; hence, there was absolutely no problem. We have a good relationship with the health facility." (Head of family with PLHIV 012)

**Table 1. Socio demographic characteristics of study participants.**

| Characteristic | Frequency | Percentage |
|---|---|---|
| *Ages in years* | | |
| 18–27 | 6 | 18.2 |
| 28–37 | 10 | 30.3 |
| 38–47 | 7 | 21.2 |
| 48–57 | 7 | 21.2 |
| 58–67 | 1 | 3.0 |
| 68–87 | 2 | 6.1 |
| *Gender* | | |
| Male | 11 | 33.3 |
| Female | 22 | 66.7 |
| *Religion* | | |
| Anglican | 22 | 66.7 |
| Catholic | 6 | 18.2 |
| Muslim | 3 | 9.1 |
| Traditionalist | 2 | 6 |
| *Occupation* | | |
| Peasant Farmer | 13 | 39.4 |
| Self employed | 9 | 27.3 |
| Employed | 7 | 21.2 |
| Unemployed | 4 | 12.1 |
| *Level of education* | | |
| Uneducated | 3 | 9.1 |
| Primary | 11 | 33.3 |
| Secondary | 9 | 27.3 |
| Tertiary | 6 | 30.3 |

*"I used to fall sick most of the time. I developed a thought that those who were already infected with HIV could have transmitted the infection to me; when I tested, they told me that I was infected with HIV . . .. They put me on treatment immediately as if the HIV had advanced'* (Expert client 1).

*Testing services.* HTS were available at the health center and in the villages through community outreaches, mainly by non-governmental organizations.

"*Like when somebody comes, we first do pre-test counseling and then we send them to the laboratory; when they turn positive, we do post-test counseling . . . . . .during post-test counseling, we disclose the result and we check someone's understanding and willingness* (Linkage Facilitator 2)

*"The health workers came home (sometimes they come house to house). They tested my mother who was sick and told me that she was HIV positive, and since then, she started HIV care from Kinoni HCIV. She has been on care and treatment for approximately five years."* (Head of family with PLHIV 4)

*Counselling services.* Counseling was one of the services readily available to newly diagnosed clients. This was provided before and after testing, and in addition, it was provided to those already in care.

*"That is how we do it. Now all of them, both from the villages brought by village health teams (community health workers) and those who have come by themselves, must undergo counseling; we carry out pre-test counseling, they consent and accept that they have agreed to test for HIV."* (Linkage facilitator 1)

*"We normally conduct counseling as in adherence and everything almost every visit till someone takes a year to feel that they are ready to take it by themselves.* (Health worker 1)

*Treatment services.* Those who test positive are initiated on treatment after counselling and drugs are available.

*"I got to know it when she came for antenatal care for her 1ˢᵗ pregnancy. You know, it is mandatory to be tested for HIV, and she tested positive for HIV. . . . . . . . . . . . . .. When the baby was born, we kept coming here and the baby tested for approximately three times, and all tests were negative. The drugs were started slowly and slowly. She had a second pregnancy. We also tested the baby, and the patient was found to be negative.* (Head of family with PLHIV 3)

*". . .the drugs are there and available; she has never missed to receive the drugs."* (Head of family with PLHIV 7)

*Follow-up and support.* Participants reported that clients are observed on follow-up and offered support by linkage facilitators.

*"We observe you through a follow-up, we get your details including your address and telephone number because if you fail to come the next day with the partner, we call you and try to give advice and encourage you to come for linkage, but majorly, we encourage test and treat."* (Linkage facilitator 2)

*"Because we even do community counseling. If it fails, we observe you through a follow-up to the community. We link up with the PLHIV network. . . . . . . . . the counseling part and the package we give, we should not become tired, keep talking, and counsel them to disclose. Never give up something even after trying for three or four times."* (Health worker 1)

*Satisfaction with the services received.* Most participants expressed their satisfaction with the services received from the health center, as reported in some of the interviews.

*"At the facility, we had no problem because the health workers acted fast. The health workers are welcoming, drugs are available, and there are counselors; hence, there was absolutely no problem. We have a good relationship with the health facility."* (Head of family with PLHIV 12)

*"There is someone (health worker) to direct you. After getting you from that room where you are tested, you will be accompanied by other subsequent rooms."* (Expert client 5)

**Theme 2: Barriers to linkage to HIV care.**    Barriers to linkage to HIV care were perceived by participants to be experienced at different levels: individual, family, health facility, and community environment.

*Individual barriers to linkage to care.* Most participants described individual barriers that hindered them from being linked to HIV care, including low socioeconomic status, high transport cost, fear of adverse effects of drugs, fear of broken relationships, denial of positive results, and denial of treatment. This is illustrated in the following excerpts:

*". . .the family has a poor economic status (low income); we noticed this when we visited their home and tested them; their viral load remained very high because this patient did not have food, which made the patient not to take the drug properly."* (Linkage facilitator 4)

*". . . they stay only in a small plot of land, which is not enough to grow many crops. . . . . .Many of them walk to the facility to pick drugs under fasting"* (Health worker 7)

*"Transport is the main challenge . . . .. coming to collect the drug costs about 4,000 Uganda shilling (US$1). . . .Sometimes, when she is needed at the facility, may be to take off a blood sample, and you find she cannot be on a motorcycle, it becomes costly to get a car, and this disturbs her that she feels like giving up."* (Head of family PLHIV 4)

*"When I was put on drugs, I took them by hiding because I never wanted my wife to know. If she knows that I am the one who has infected her, she may feel angry and decide to separate with me and my children, and I may suffer when she goes."* (Expert client 003)

*". . .. Some deny their results saying that the machine is not of good quality, especially in people find it hard to deal with and to follow. . .At the community level, when this patient is tested positive and you refer them to the facility, they may take a long time to come."*(Health worker, 5)

*". . . . .patients can still be in denial and continue abusing drugs and some come when they are drunk; those people are not ready to take ARVs. . ."* (Linkage facilitator 4)

*Health facility barriers to linkage to care.* Relating to the health facility services, some participants reported that they had to wait for a long time to access services, and some staff had negative attitudes toward patients. The stock outs of some drugs were also mentioned by many participants.

*. . .. I reached here at around 9:00 am, and now it is 1:00 pm, but they haven't worked on me. I had come with three books, but now I have received only two books; when I told them that I had a third book, no one listened, and later, I found that the book was just dumped and left there, and the responsible person went in their own business."* (Expert client 1)

*"There is a problem with time here. You come when you have not taken anything hoping to be worked on so fast, but you are delayed here and get dizziness because of hunger"* (Head of family PLHIV 7)

*". . ... They make us wait for so long to the extent that even the person you did not want to see you will see you. You will be on the line for the whole day."* (Expert client 10)

*"—availing Septrin [cotrimoxazole] at the health facility because sometimes you reach and they tell you to go and buy Septrin. Yet you do not have any single coin; then the child takes a week without taking the drug, which I think can deteriorate the health of the child."* (Head of family PLHIV 9)

*"We also have challenges in stock out of some of the drugs, and as you know, all patients who have started treatment and they are below have not yet reached 6 months and their viral loads are not assessed to see whether they are suppressing; they have to give Septrin [cotrimoxazole]. Sometimes, you find that we don't have that Septrin for children, so it's also a challenge."* (Health worker 2)

*Community barriers to linkage to care.* In the community, stigma and discrimination were the most common barriers described by the participants. They reported having had a lot of stigma, discrimination, and embarrassment while receiving HIV care services. They fear that some people will see them and start pointing fingers. This is captured in the following statements:

*People had started asking me why I am always at the heath facility, yet I did not want them to know. I am tired of these questions. This is why I left my home area and transferred to the health center. They [health workers] gave me drugs, but the problem again I got is that some people from my home are also getting drugs from here (Kinoni HCIV). I now again said 'should I kill myself or,' but later, I saw that it would not benefit me to die. But in the community where we stay, when some people know that you are HIV positive, they start running away from you (segregating you)."* (Expert client 9)

**Theme 3: Facilitators to linkage of care.** There were a number of facilitating factors mentioned by participants, and five subthemes were identified, including positive staff attitudes, access to information, follow-up support, fear of death, and support from others.

*Positive attitudes of staff.* Most participants reported that healthcare providers were friendly, receptive, and patient-centered. This is illustrated in the following excerpt:

*"When I came for drugs, the health workers were welcoming. They listened to my complaints and worked on me. . . .when I was very sick, they looked for a doctor who saw me and gave me drugs for my sickness."* (Expert client 6)

*Access to information.* The participants acknowledged always being given information as described below:

*"The child got sick for a long time, and we would move here and there looking for medical services but in vain. What opened our eyes first are people who used to teach us that when you have a child coughing too much, he/she may be having HIV."* (Head of family PLHIV 10)

*Fear of death.* After receiving a positive HIV result, fear of death was another factor that drove people to be linked to HIV care services.

*". . .It was not easy. I started thinking that if I start the drugs, I will die, but again, I thought that if I don't start the drugs, I will still die. . .. . .. I started the drugs but got disturbed with it because I had a lot of fear."* (Expert client 5)

*Support from others.* HIV-positive patients receive support from family members, the community, and humanitarian non-governmental organizations.

*"They [health workers] tell us that these drugs [antiretroviral] need to be taken when you have eaten or had a soft drink so I have to provide them, and she also has to take them in time, that is, breakfast, lunch, and supper. Even taking something after taking the drugs. Transport to bring her to a health facility is available to her. I keep reminding her to swallow the drugs. I come to pick drugs for her when she is not around so that she keeps with a constant supply."* (Head of family PLHIV 3)

*"I advise him every morning before taking the drugs to first eat something and then have a drink. I remind him every day to take the drugs without forgetting, and she also reminds me of taking mine. When she becomes pregnant, I try by all means to see that she delivers in a health facility by trained health professionals so that the child born remains healthy without HIV."* (Head of family PLHIV 7)

*"I was lucky to get some neighbor who helped me with daily milk to give to those young children so that they can grow up. Another remaining portion of milk would be put in a porridge for them to obtain the energy required to grow. Getting clothing was not easy for me until Compassion International came and assisted me by taking up these children.* (Head of family PLHIV 11)

**Theme 4: Suggestions for improving service delivery.**   Various suggestions were provided by clients and health workers, including shortening waiting time, integrating HIV care with general services, increasing staff numbers at the health center, and outreach services.
*Shortening waiting time.* Participants reported that if waiting time could be shortened, service provision would improve.

*"Health workers should improve in service provision such that when I come early, I go early but not looking at the faces of people. . .. . .. ."* (Head of family PLHIV 6)

*Integrating HIV care with other services.* Participants suggested the need to integrate HIV care with other routine services (one-stop center) to prevent stigmatization from other patients.

*". . .. In my own opinion, those people would be made to sit just like other patients when they come to get their drugs. I think this could help reduce stigmatization."* (Health worker, 2)

*Increasing staff numbers.* A large number of participants sought care at the health center, thereby imposing pressure on the limited number of staff. Increasing the number of health workers would go a long way to improve care.

*". . .. People (patients) are many, and the number of health workers should be increased so that they work faster and we take a shorter time, especially when we come. . ."* (Head of family PLHIV 6)

*Outreach services.* Outreach services for HIV care were another area where participants needed more support for improved services. This was compared to other services, such as immunization, which are offered in a similar manner.

*". . . should improve on HIV outreaches like they do for immunization. . .they can organize a place where they can meet and share certain things and we give them the necessary services that are required."* (Linkage facilitator 1)

## Discussion

The study aimed to identify the perspectives of health care providers and caretakers of PLHIV on existing measures in place that promote linkage to care, explore facilitators and barriers to linkage to care for clients diagnosed with HIV at a rural health center in South Western Uganda. This study identified existing measures in place to promote linkage to care, explored facilitators and barriers to linkage to care, and collected suggestions for improving the linkage of clients diagnosed with HIV to care in a rural health center in South Western Uganda. Existing measures in place to promote linkage of clients diagnosed with HIV to care were through the availability of HIV care services, including testing, counseling, treatment, and support services. Barriers to the linkage of HIV care were categorized as individual, health facility, and community. Individual barriers were due to socioeconomic status, high transport cost, fear of adverse effects of drugs, fear of broken relationships, denial of treatment, and drug abuse, while health facility barriers were negative staff attitudes and long waiting times. On the contrary, community barriers were mostly due to stigma experienced by HIV clients, which, in some cases, result in discrimination by community members. In this study, a number of facilitators to linkage to HIV care were identified, including positive staff attitudes, access to information, follow-up support, and fear of death. The study also offered suggestions for improving service delivery at the health facility for clients with HIV, including shortening waiting time, integrating HIV care with general services, increasing staff numbers, and outreach services to communities.

The availability of services identified in this study, such as HIV counseling and testing, initiation of ARVs, referral, and follow-up support is an indication that the sustained support by various agencies over a number of years has led to improvements in the provision of HIV care services. Similar findings were also identified in a study conducted in Mbeya Tanzania that reported the availability of HIV care services, including a referral system, in the health facility, which was effective in linking PLHIV into care [13]. Other authors have reported that, to effect linkage among health facilities, strategies for immediate referral following HIV diagnosis are of great importance [27].

Despite the "test and treat" strategy as recommended by the UNAIDS, barriers to linkage to care remain a big hindrance, leading to loss of care for some patients [6]. Loss of care is associated with reduced viral suppression, increased HIV morbidity, and a high risk of transmission [28, 29]. Key barriers to linkage to care identified in our study, such as low socioeconomic status, negative attitudes of staff, and stigma, have been reported in several studies conducted in Uganda and other sub-Saharan African countries [30–32]. Regarding stigma, similar situations have been noted in a study conducted in South Africa, Lesotho Malawi, Swaziland, and Tanzania, leading to patients shunning care or missing out on medications because of perceived

stigma [33, 34]. This might indicate that all HIV-positive individuals, irrespective of their location, experience considerable stigma [35]. Studies conducted elsewhere in Malawi and China highlighted negative attitudes of staff as one of the barriers to linkage to care [36, 37]. This affects the uptake of HIV services from facilities that negatively influence patient satisfaction. Transport-related costs were another hindrance to linkage to care. Such constraints can affect patients' decisions regarding linkage to HIV care. A study conducted in South Africa also reported on how transport for patients can be underscored when considering referral to linkage to care [31].

Conversely, our study found several factors that facilitated linkage to care. These include positive attitudes of staff, access to information, support, and fear of death. These factors seem to have contributed to the current level of HIV linkage to care at health facilities. Other studies found that peer educators, mass communication, and social support from peers were important in promoting linkage to HIV care [31, 38]. Similar to our study findings, other researchers found that health care providers' relationships with patients and provision of adequate information to patients promoted patients' linkage to care [39, 40]. Family and community support identified in our study was reported to positively impact linkage to care, which promotes acceptance and thus limits self-stigma. This is in line with the findings of previous studies conducted in Tanzania and Botswana that reported that support from peers and community members were major facilitators in linkage to care for PLHIV peers escorted HIV patients to care points, and the majority of the patients have access to care [31, 41].

Linkage to care for persons tested positive for HIV is a process that is influenced by several factors. It starts with the individual who receives the positive test result, and according to the health belief model [24, 25] the decision to take the next course of action depends on how the person perceives the severity of the disease, health motivation, perceived benefits, and barriers. Although the individual may be inclined to continue in the process of linkage to care, they are confronted with structural factors in the community, health system levels as spelt out in the ecological model [26]. In this study, the health care providers and caretakers of PLHIV were able to bring out the community and health system factors that hinder or promote linkage to care. As the HIV pandemic transitions to a chronic disease, new models are needed that will promote linkage to care amidst emerging competing priorities [42].

Suggestions for improving linkage to care were another theme that stemmed from the study. Participants suggested that to improve services, reducing waiting time and integration of services were crucial. We term this as "a one-stop center" where a patient could receive all the needed services concurrently. This would prevent stigmatization from other patients and promote patient satisfaction. Studies on integration of HIV care with other health services have shown positive outcomes in Kenya, Rwanda, Malawi, and South Africa [43–46] Intensified outreach services for HIV care was another important service that we identified in our study. It is envisaged that to improve linkage to care, services should be brought nearer to the people in the community, which reduces transport costs and prevent defaulting. Our study finding is in line with those of another study conducted in Mbeya, Tanzania, which reported that more people access HIV services at outreach sites than at health facilities [13]. Likewise, other authors showed that outreach services for HIV care were vital in improving linkage to care and emphasized that to improve linkage to care, health systems' connection with communities should be strengthened to include collaborations [47, 48].

## Strengths and limitations of the study

The purposively selected individuals drawn from expert clients, heads of families with PLHIV, linkage facilitators, and health workers included a mix of various types of participants who

provided responses on the linkage of HIV clients to care from different perspectives. The use of in-depth interviews as a data collection method enabled the study team to collect more detailed information on the participants' perspectives on linkage to care for persons diagnosed with HIV than would have been obtained using other data collection methods.

The main limitation of this study was the failure to obtain perspectives from PLHIV who are not engaged in service provision. There were limitations on person movements due to the coronavirus disease (COVID-19) pandemic lockdown restrictions in Uganda at the time of the study (July to August 2020). The study could not receive the views of clients that were linked to care for a brief period and thereafter defaulted. It is recommended that future studies could take a longitudinal approach in which clients are observed on follow-up starting from the HIV-testing stage. Conducting the study during the COVID-19 pandemic was disruptive, as participants were uncomfortable with some of the standard operating procedures/guidelines in place at the health center. However, this was overcome by educating them on how to put on the masks, handwashing, and the importance of practicing social distancing. The design of the study is qualitative, and the findings cannot be generalized, which is a limitation.

## Conclusion

Linkage to HIV care is important to achieve the UNAIDS 90-90-90 target. We found that access to linkage to care for patients diagnosed with HIV are positively and negatively influenced at the individual, community, and health facility levels. However, integration of HIV services and intensifying outreaches are key to improving linkage to care. Policy makers and implementers should consider the issues identified in this study to make the test and treatment strategy more successful.

## Supporting information

**S1 File. Interview guides in English.**
(DOCX)

**S2 File. Consolidated criteria for reporting qualitative studies (COREQ) checklist.**
(DOCX)

**S3 File. Original transcripts.**
(DOCX)

**S1 Table. Data coding summary showing codes, subthemes and themes.**
(DOCX)

## Acknowledgments

### Participants

We thank those who volunteered to participate in this research and the district local leaders who granted us permission to conduct the study in the area.

### Kinoni health center management

We thank the facility managers for granting us permission to conduct the study and to provide a conducive environment during the study.

## Author Contributions

**Conceptualization:** Florence Akello, Doreen Kagina Twongyeirwe, David Opio, Juliet Aceng, Jane Kasozi Namagga, Jerome Kahuma Kabakyenga.

**Data curation:** Mark Opio, Florence Akello, Doreen Kagina Twongyeirwe, David Opio, Juliet Aceng, Jane Kasozi Namagga, Jerome Kahuma Kabakyenga.

**Formal analysis:** Mark Opio, Jane Kasozi Namagga, Jerome Kahuma Kabakyenga.

**Funding acquisition:** Mark Opio, Florence Akello, Doreen Kagina Twongyeirwe, David Opio, Juliet Aceng, Jane Kasozi Namagga.

**Investigation:** Mark Opio, Florence Akello, Jane Kasozi Namagga, Jerome Kahuma Kabakyenga.

**Methodology:** Mark Opio, Florence Akello, Doreen Kagina Twongyeirwe, David Opio, Juliet Aceng, Jane Kasozi Namagga, Jerome Kahuma Kabakyenga.

**Project administration:** Mark Opio, David Opio, Juliet Aceng, Jane Kasozi Namagga.

**Resources:** Mark Opio.

**Software:** Jerome Kahuma Kabakyenga.

**Supervision:** Mark Opio, Jane Kasozi Namagga, Jerome Kahuma Kabakyenga.

**Validation:** Jane Kasozi Namagga, Jerome Kahuma Kabakyenga.

**Visualization:** Mark Opio, Florence Akello, Jane Kasozi Namagga, Jerome Kahuma Kabakyenga.

**Writing – original draft:** Mark Opio, David Opio, Juliet Aceng, Jane Kasozi Namagga, Jerome Kahuma Kabakyenga.

**Writing – review & editing:** Mark Opio, Florence Akello, Doreen Kagina Twongyeirwe, David Opio, Juliet Aceng, Jane Kasozi Namagga, Jerome Kahuma Kabakyenga.

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
