## [Decision Letter · Decision Letter 0]

28 Jul 2021

PONE-D-21-00344

Improving Linkage to Care for Patients Diagnosed with HIV: A Qualitative Study at a Rural Health Centre in Southwestern Uganda

PLOS ONE

Dear Dr. Kabakyenga,

Thank you for submitting your manuscript to PLOS ONE. After careful consideration, we feel that it has merit but does not fully meet PLOS ONE’s publication criteria as it currently stands. Therefore, we invite you to submit a revised version of the manuscript that addresses the points raised during the review process.

Reviewer 1:

**1- General comments on the article and its content **

The concept *‘’ Linkage to care’’* discussed within this article is not clear and consistent along the article. Some time, it refers to linkage to ART; sometimes, it refers to linkage to ART, retention up to access to VL.... In brief, it’s not very clear whether what was studied is linkage to ART treatment or linkage to care 1-month or 3 -6 months, etc. following HIV diagnosis?The article is discussing about facilitators and barriers to improve linkage to care of HIV positive clients from the providers perspective. It’s true that key respondents are Expert clients and linkage facilitators but, for the provider perspective not the client’s perspective; That precision is missing along the article

**2- Comments on the Introduction section: **

Example 1: Line 79 to 81. You have a sentence like: *‘’ **Similarly, a country progress report on access to ART services in one of the 80 health facilities in Kabale District in southwestern Uganda, reported that linkage to care was at 81 67% [8].’’*

The Sentence started with report on access to ART servicesConcluded with a percentage on linkage to care

Example 2: Line 83 to 85. The sentence is about linkage to care but compared to the second 90 (UNAIDS-goal which is about linkage to ART treatment)

These 2 examples are illustration of mix-up between linkage to care and linkage to ART treatment observed all-over the article

**2- Comments on the Methods section: **

Line 122. The sentence says: *‘’ **The interview guides were translated and back translated to maintain the flow of the information** ‘’. *

Translation during interview is very importantThe authors should share more details about the translators’ characteristics (certified or not; experience in translation in health sector; etc..) and approval process of translationThis can influence enough the in-depth interview, the understanding of the question and the answers

**3- Comments on the Results section: **

Participants were providers, not really patients using the services; not all (like health care workers; etc..)Difficult understand some of the conclusion like line 160 to 161 ‘*’** Most of the participants reported receiving good services at the health facility.’’ *Some of the participants being health care workersLine 165 to 172 : it’s a linkage facilitator answer and perspective. We should have love having the client perspective on all those aspect to better understand acceptance or refusalLine 173 to 176 : The head of familly is not sharing his personal experience but the experience of the Mother..This is the caregiver perspective ; not really the patient perspectiveLine 190 to 192. The expert client enrolled in the study for the perspective of provider (see description in Methods section) is interviewed for the perspective of patient ; nothing is clear whether, the experience of client described is at one of the study facility or ElsewhereIndividual barriers are from the providers perspective not the patient perspective [see Line 228 (feedback from linkage facilitator) and line 230 (feedback from Health care worker)]We are missing view of patients missing in care ( defaulters or LTFU) and re-engaging later

**4- Comments on the Discussion section: **

The construction of themes using both Health belief Models and Ecological Models was not really discussed to help better understand the resultsThe health belief model focus on individual factors; and in this study, the provider perspective was considered not the patient perspective; in particular patients re-engaging in care who are in better place to inform us about individual barriers. 

Reviewer 2: 

**Introduction**

The introduction is relatively short with minimal information on what is already known about linkage to care, there are so many articles in this topic the authors may wish to add more literature.

It is always nice  to start with information about the magnitude of the problem  worldwide,  in Africa or Sub Saharan Africa,  East Africa and then  in Uganda,  below are some of the articles you may wish to include for East Africa, studies conducted in  Kenya, Tanzania etc

You may wish to start with line 72: Globally, in 2019, only 67% of HIV positive clients were linked to HIV care .....Then follow with the definition and the effects of failure or poor linkage....

**Line 76-78:** The results showed that only 53% of the HIV positive 77 clients were linked to care within one month after testing while 14.5% never returned for follow 78 up [7]

This statement does not fit here, unless the authors will also address the issues of time to linkage and retention issues, if not I suggest dropping it.

Some of the articles

Ruzagira et al., 2017: Linkage to HIV care ..... in sub-Saharan Africa: a systematic review: https://onlinelibrary.wiley.com/doi/10.1111/tmi.12888Sanga, E.S. *et al.*, 2019: Understanding factors influencing linkage to HIV care .... Mbeya- Tanzania: qualitative findings of a mixed methods study. *BMC Public Health* **19, **383 (2019). https://doi.org/10.1186/s12889-019-6691-7Kegoli S et al., 2019: increasing linkage to HIV care and ... HIV patients at 14 EDARP health facilities in Nairobi, Kenya. https://www.ajol.info/index.php/eamj/article/view/198131

**Methodology**

About sampling, the authors said it was purposive sampling, i would like to know if the did consider factors like age, sex, occupation, education level and may be distance from the health facility/ HIC clinic offering HIV services to increase the diversity in respondents?

**Line 122 and 127:** The authors reported that, the interview guides were translated and back translated ... it would be nice to also mention the languages the guide were translated

 to /from

**Line 131-136****...**As the first step of data analysis, all 132 audio recordings were transcribed... It could just be okay to say XX and XX cross-checked or verified the transcription and translation of data before analysis

**Line137-142:** the authors mentioned the Health belief and ecological models, i think it would be nice to briefly mention their components and probably link it to how the used the model to analyze the study findings

**Result section**

**Line 150-157:** Socio demographic characteristics of participants- The authors may wish to put these in a table, I see more qualitative papers nowadays have these presented in tables,

Just after the Participants characteristics, I suggest you start with an overview of the study, i.e. six /five/two/ theme related to .... *Study topic* were identified and categorized in the following sub headings. You only mention in this paragraph and then you start explaining the details one by one. This guides the reader to know beforehand that okay there ... themes/sub-heading to come

**Line 158-161:** Availability of services; At least one quotation is needed to back up the explanations in this paragraph

Also when you have long quotations is good to indent them

**Discussion**

In this section is good to start by mentioning what was the study looking for and then what are the main findings in your study then you continue with explaining the importance and implications of the finding and how are their comparable to other r similar studies. I noted some repetition of results in the discussion; is okay to refer to the result, not to report again

**Conclusion:**

 Excellent no comments

We look forward to receiving your revised manuscript.

Kind regards,

Habakkuk A. Yumo, MD, MSc, PhD

Academic Editor

PLOS ONE

Additional Editor Comments (if provided):

Dear Dr. xxxxxxxx

Thank you for submitting your manuscript to PLOS ONE. After careful consideration, we feel that it has merit but does not fully meet PLOS ONE’s publication criteria as it currently stands. Therefore, we invite you to submit a revised version of the manuscript that addresses the points raised during the review process.

The manuscript has been evaluated

by two reviewers, and their comments are available below.

The reviewers have raised a number of concerns that need attention. They request clarity on the definition of some concepts, the methodological aspects of the study and on the presentation of some findings in results and discussion sections.

Could you please revise the manuscript to carefully address the concerns raised?

In addition, please kindly ensure your revising style is consistent with PLOS ONE guideline (https://journals.plos.org/plosone/s/submission-guidelines).

Journal Requirements:

A clean copy of the edited manuscript (uploaded as the new *manuscript* file).

3. Thank you for submitting an English version of your questionnaire. Please also include a copy of the questionnaire in the original language, as Supporting Information, or include a citation if it has been published previously

Reviewers' comments:

Reviewer's Responses to Questions

**Comments to the Author**

1. Is the manuscript technically sound, and do the data support the conclusions?

Reviewer #1: Partly

Reviewer #2: Yes

2. Has the statistical analysis been performed appropriately and rigorously? 

Reviewer #1: Yes

Reviewer #2: N/A

3. Have the authors made all data underlying the findings in their manuscript fully available?

Reviewer #1: No

Reviewer #2: No

4. Is the manuscript presented in an intelligible fashion and written in standard English?

Reviewer #1: No

Reviewer #2: Yes

6. PLOS authors have the option to publish the peer review history of their article (what does this mean?). If published, this will include your full peer review and any attached files.

Reviewer #1: No

Reviewer #2: No

---

## [Author Response · Author response to Decision Letter 0]

23 Oct 2021

RESPONSES TO REVIEWERS’ COMMENTS on the study: Improving Linkage to Care for Patients Diagnosed with HIV: A Qualitative Study at a Rural Health Centre in Southwestern Uganda 

Reviewers Comments Responses 

R1 1-General comments on the article and its content

 We welcome the comments as they are aimed at making our manuscript better and more informative.

R1 The concept ‘’ Linkage to care’’ discussed within this article is not clear and consistent along the article. Some time, it refers to linkage to ART; sometimes, it refers to linkage to ART, retention up to access to VL.... In brief, it’s not very clear whether what was studied is linkage to ART treatment or linkage to care 1-month or 3 -6 months, etc. following HIV diagnosis? Linkage to care for HIV positive patients is a process of enrolling and ensuring all HIV positive patients have access to Antiretroviral Therapy (ART) and care services. Linkage to care starts with HIV testing services and is followed by post-test counselling, clinical evaluation, Anti-retroviral therapy (ART) initiation, early support up to the first follow-up visit. Initiation on ART is a key activity in linkage to care but has to be supported by other activities before and after [lines 71-87].

R1 The article is discussing about facilitators and barriers to improve linkage to care of HIV positive clients from the provider’s perspective. It’s true that key respondents are Expert clients and linkage facilitators but, for the provider perspective not the client’s perspective; that precision is missing along the article

 We agree with the reviewer that the manuscript is majorly on facilitators and barriers to linkage to care. The expert clients, linkage facilitators and health workers can be taken as health care providers while the heads of families with PLHIV are caretakers. The correction has been made [lines 134-135].

R1 2-Comments on the Introduction Section 

R1 Example 1: Line 79 to 81.you have a sentence like “similarly, a country progress report on access to ART services in one of the 80 health facilities in kabala District in southwestern Uganda, reported that linkage to care was at 67% [8]. 

• The Sentence started with report on access to ART services

• Concluded with a percentage on linkage to care We accept there was an error. We have thus corrected “ART” to “HIV” care services [line 106]. 

 R1 Example 2: Line 83 to 85. The sentence is about linkage to care but compared to the second 90 (UNAIDS-goal which is about linkage to ART treatment)

These 2 examples are illustration of mix-up between linkage to care and linkage to ART treatment observed all-over the Article

 Thank you for the observation. 

As already indicated in the description of linkage to care for those diagnosed with HIV has several components and linkage to ART treatment is the most prominent easily measurable.

 2-Comments on the Methodology 

R1 Line 122. The sentence says: ‘’ The interview guides were translated and back translated to maintain the flow of the information ‘’.

Translation during interview is very important

The authors should share more details about the translators’ characteristics (certified or not; experience in translation in health sector; etc..) and approval process of translation

This can influence enough the in-depth interview, the understanding of the question and the answers The interview guides were designed in English. The interview guides for health workers and linkage facilitators were administered in English while the guides for Expert clients and heads of families with PLHIV were first translated to the local language and then back translated to check whether the flow of information was maintained. The translation was carried out by persons certified/recognized by the local institutional ethics review board lines [141-158]

 3-Comments on the Results 

R1 Participants were providers, not really patients using the services; not all (like health care workers; etc.)

Difficult understand some of the conclusion like line 160 to 161 ‘’ Most of the participants reported receiving good services at the health facility.’’ Some of the participants being health care workers The participants were a mix of providers (health workers, linkage facilitators) and caretakers – service users (heads of families with PLHIV). The expert clients were both peer educator/supporters and at the same time service users.

R1 Line 190 to 192. The expert client enrolled in the study for the perspective of provider (see description in Methods section) is interviewed for the perspective of patient; nothing is clear whether, the experience of client described is at one of the study facility.

 You are right.

This excerpt has been deleted and has been replaced with an appropriate one.

R1 Elsewhere Individual barriers are from the providers perspective not the patient perspective [see Line 228 (feedback from linkage facilitator) and line 230 (feedback from Health care worker)] 

We are missing view of patients missing in care ( defaulters or LTFU) and re-engaging later

 It is true the individual barriers are from the perspectives of the providers and some extent from caretakers. It is the expert clients’ views that can be taken to come from the patients. 

 4-Comments on the Discussions 

R1 • The construction of themes using both Health belief Models and Ecological Models was not really discussed to help better understand the results. 

• The health belief model focus on individual factors; and in this study, the provider perspective was considered not the patient perspective; in particular patients re-engaging in care who are in better place to inform us about individual barriers. The health belief model and ecological models have been catered for in the discussion [lines 448-457].

 REVIEWER 2 

 1.Introduction 

R2 The introduction is relatively short with minimal information on what is already known about linkage to care. Thank you for that observation.

Some more text has been introduced in the introduction. 

R2 The magnitude of the problem worldwide, in Africa or Sub Saharan Africa, East Africa and then in Uganda, below are some of the articles you may wish to include for East Africa, studies conducted in Kenya, Tanzania etc 

 The magnitude-globally, regionally and locally in Uganda has been captured [lines 63-70 & 88-105].

R2 Line 76-78: The results showed that only 53% of the HIV positive 77 clients were linked to care within one month after testing while 14.5% never returned for follow 78 up [7]

This statement does not fit here, unless the authors will also address the issues of time to linkage and retention issues, if not I suggest dropping it. Thank you for the observation.

We took linkage to care to be a process that starts with the first test that turns out to be positive up to the period after about one month. This is now reflected in lines 71-76. 

 2. Methodology 

R2 About sampling, the authors said it was purposive sampling, i would like to know if they did consider factors like age, sex, occupation, education level and may be distance from the health facility/ HIC clinic offering HIV services to increase the diversity in respondents? Thank you.

The factors were not considered, since it wasn’t defined; although the diversity was met according to socio-demographics, and the different groups of participants (expert clients, linkage facilitators, health workers, heads of families with PLHIV). 

R2 Line 122 and 127: The authors reported that, the interview guides were translated and back translated ... it would be nice to also mention the languages the guide were translated to/from

 Comment has been well received.

This has now been corrected [lines 151-154].

R2 Line 131-136...As the first step of data analysis, all [132] 33 audio recordings were transcribed... It could just be okay to say XX and XX cross-checked or verified the transcription and translation of data before analysis The interviews were 33.

Transcribed and translated data was crosschecked by researchers OM, TKD, AJ and AF, although we prefer putting research team.

R2 Line137-142: the authors mentioned the Health belief and ecological models, i think it would be nice to briefly mention their components and probably link it to how the used the model to analyze the study findings

 Thank you for the observation. This has been taken care of under Lines 175-180

Generation of themes were guided by two theoretical constructs derived from the Health Belief Model that included Perceived susceptibility, severity, Health Motivation, Perceived benefits, barriers, and Cue to action. While ecological Model was composed of the health system, individual and community structural levels

 Results Section 

R2 Line 150-157: Socio demographic characteristics of participants- The authors may wish to put these in a table, I see more qualitative papers nowadays have these presented in tables,

Just after the Participants characteristics, I suggest you start with an overview of the study, i.e. six /five/two/ theme related to ... Study topic were identified and categorized in the following sub headings. You only mention in this paragraph and then you start explaining the details one by one. This guides the reader to know beforehand that okay there ... themes/sub-heading to come Thank you for the comment.

Table 1 (sociodemographic characteristics of participants) has been introduced in the manuscript.

We identified four themes that were categorized as availability of services, barriers to linkage of care, facilitators to linkage of care and improving service delivery as described [lines 200-204].

R2 

 Line 158-161: Availability of services; At least one quotation is needed to back up the explanations in this paragraph .Also when you have long quotations is good to indent them

 Thank you for the comment. 

We have supported the theme with two quotes [lines 208-213] 

 Discussion 

R2 In this section is good to start by mentioning what was the study looking for and then what are the main findings in your study then you continue with explaining the importance and implications of the finding and how are their comparable to other r similar studies. I noted some repetition of results in the discussion; is okay to refer to the result, not to report again Thank you for the comment. 

In paragraph 1 in the discussion section, we tried to summarise the results for the reader to be updated of the key findings. Subsequent paragraphs are on the discussion of the individual themes and comparing our findings with what is already in the literature.

 Conclusion 

R2 Excellent no comments No comments

---

## [Decision Letter · Decision Letter 1]

31 Jan 2022

Perspectives on linkage to care for patients diagnosed with HIV: A qualitative study at a rural health center in South Western Uganda

PONE-D-21-00344R1

Dear Dr. Kabakyenga,

We’re pleased to inform you that your manuscript has been judged scientifically suitable for publication and will be formally accepted for publication once it meets all outstanding technical requirements.

Kind regards,

Habakkuk A. Yumo, MD, MSc, PhD

Academic Editor

PLOS ONE

Additional Editor Comments (optional):

Reviewers' comments:

Reviewer's Responses to Questions

**Comments to the Author**

1. If the authors have adequately addressed your comments raised in a previous round of review and you feel that this manuscript is now acceptable for publication, you may indicate that here to bypass the “Comments to the Author” section, enter your conflict of interest statement in the “Confidential to Editor” section, and submit your "Accept" recommendation.

Reviewer #2: All comments have been addressed

2. Is the manuscript technically sound, and do the data support the conclusions?

Reviewer #2: Yes

3. Has the statistical analysis been performed appropriately and rigorously? 

Reviewer #2: Yes

4. Have the authors made all data underlying the findings in their manuscript fully available?

Reviewer #2: No

5. Is the manuscript presented in an intelligible fashion and written in standard English?

Reviewer #2: Yes

6. Review Comments to the Author

Reviewer #2: The authors have addressed all comments to my satisfaction , i have no further comment, I recommend the manuscript to be published in this journal, unless the other reviewers have more comments

7. PLOS authors have the option to publish the peer review history of their article (what does this mean?). If published, this will include your full peer review and any attached files.

Reviewer #2: **Yes: **Dr Erica Samson Sanga- The National institute for Medical Research- Mwanza, Tanzania

---

## [Editor Report · Acceptance letter]

24 Feb 2022

PONE-D-21-00344R1 

Perspectives on linkage to care for patients diagnosed with HIV: A qualitative study at a rural health center in South Western Uganda 

Dear Dr. Kabakyenga:

I'm pleased to inform you that your manuscript has been deemed suitable for publication in PLOS ONE. Congratulations! Your manuscript is now with our production department. 

Kind regards, 

on behalf of

Dr. Habakkuk A. Yumo 

Academic Editor

PLOS ONE